# Intra-Articular Injection of Autologous Micro-Fragmented Adipose Tissue for the Treatment of Knee Osteoarthritis: A Prospective Interventional Study

**DOI:** 10.3390/jpm13030504

**Published:** 2023-03-10

**Authors:** Yang Yu, Qunshan Lu, Songlin Li, Mingxing Liu, Houyi Sun, Lei Li, Kaifei Han, Peilai Liu

**Affiliations:** 1Department of Orthopaedics, Qilu Hospital of Shandong University, Jinan 250102, China; 2Cheeloo College of Medicine, Shandong University, Jinan 250012, China; 3Chinese Academy of Medical Sciences and Peking Union Medical College, Beijing 100730, China; 4Boshan District Hospital of Traditional Chinese Medicine, Zibo 255200, China

**Keywords:** knee osteoarthritis, micro-fragmented adipose tissue, cartilage repair, mesenchymal stem cells

## Abstract

Background: To investigate the efficacy and safety of autologous micro-fragmented adipose tissue (MF-AT) for improving joint function and cartilage repair in patients with knee osteoarthritis. Methods: From March 2019 to December 2020, 20 subjects (40 knees) between 50 and 65 years old suffering from knee osteoarthritis were enrolled in the study and administered a single injection of autologous MF-A. The data of all patients were prospectively collected. The Western Ontario and McMaster Universities Osteoarthritis Index (WOMAC), knee society score (KSS), hospital for special surgery (HSS) score, visual analogue score (VAS) pain score, changes in cartilage Recht grade on magnetic resonance imaging (MRI) and adverse events were analyzed before and 3, 6, 9, 12 and 18 months after injection. Results: The WOMAC, VAS, KSS and HSS scores at 3, 6, 9, 12 and 18 months after injection were improved compared with those before injection (*p* < 0.05). There was no significant difference in WOMAC scores between 9 and 12 months after injection (*p* > 0.05), but the WOMAC score 18 months after injection was worse than that at the last follow-up (*p* < 0.05). The VAS, KSS and HSS scores 9, 12 and 18 months after injection were worse than those at the last follow-up (*p* < 0.05). The Recht score improvement rate was 25%. No adverse events occurred during the follow-up. Conclusions: Autologous MF-AT improves knee function and relieves pain with no adverse events. However, the improved knee function was not sustained, with the best results occurring 9–12 months after injection and the cartilage regeneration remaining to be investigated.

## 1. Introduction

Osteoarthritis (OA) is a kind of joint degenerative disease caused by many factors and leads to irreversible cartilage damage, subchondral osteosclerosis and synovitis [1]. It is caused by degeneration of articular cartilage and subchondral bone. According to a previous survey, approximately 9.6% of men and 18% of women over the age of 60 years suffer from symptomatic OA worldwide [2]. Currently, many treatment modalities for knee OA, such as lifestyle modification and pharmaceuticals, are advocated [3]. However, advanced knee OA eventually requires joint surgery as the disease progresses [2,4,5]. It is still unknown how to reverse the progression of osteoarthritis of the knee.

Micro-fragmented adipose tissue (MF-AT) is composed of three-dimensional biological scaffolds and colonies of cells in which the three-dimensional biological scaffolds are composed of collagen and connective tissue and microvascular networks, and the clustered cells consist of pericytes, adipocytes and MSCs, among others [6]. Mesenchymal stem cells (MSCs) are pluripotent stem cells that have the potential to self-renew and to multidifferentiate in a variety of tissues in the body [7]. With characteristics of easy expansion, self-renewal, multidiversification and low immunogenicity, they are usually derived from adipose tissue, umbilical cord blood or bone marrow [8,9,10,11]. Studies have shown that intra-articular injections of MSCs can treat cartilage in the knee [11]. The adipose mesenchymal stem cells (AD-MSCs) contained in autologous MF-AT are one of the many MSCs that have the ability to differentiate into articular chondrocytes. In addition, more than 100 kinds of cytokines have been detected to endow MF-AT with strong antibacterial, anti-inflammatory and antiapoptotic properties that promote vascular regeneration and tissue repair [12]. Although previous studies have confirmed the effectiveness of intra-articular injection of autologous MF-AT for knee OA, the effective time and administration interval of MF-AT remain to be explored.

Therefore, we conducted a prospective interventional study to evaluate the change in clinical effect over time of intra-articular injection of autologous MF-AT and their effect on cartilage repair in patients with knee OA. The purpose of this study is as follows: 1. the effect and duration of autologous MF-AT on functional recovery of the knee joint; 2. the effect on the repair of cartilage; and 3. its safety.

## 2. Materials and Methods

### 2.1. Study Design

The study is a single-center, interventional, prospective, interventional study (NCT 03956719). The study protocol was approved by the Ethics Committee of Qilu Hospital of Shandong University (KYLL-2018-023), and all participants provided written informed consent.

### 2.2. Patient Selection

From March 2019 to December 2020, patients were recruited from an outpatient clinic because of symptomatic knee osteoarthritis, and the first eligible outpatient per month was included. The inclusion criteria of the study were as follows: (1) patients aged between 50 and 65 years, and an American Society of Anesthesiologists (ASA) score from grade one to grade three; (2) patients with bilateral knee OA; and (3) patients with a Recht grade 1–3 on MRI. The exclusion criteria were as follows: (1) patients with acute joint injury, knee joint tuberculosis, tumor or rheumatic diseases; (2) pregnant or lactating women; (3) patients with an allergic constitution; (4) patients who underwent other knee surgery performed within 6 months; (5) those who had incomplete data affecting the efficacy or safety judgment; or (6) subjects who could not understand and voluntarily sign the written informed consent or comply with the research protocol and visitation process.

### 2.3. Surgical Procedure

The procedure was performed at the same place by the same specialized physician who was not involved in any of the evaluations of the participants. Autologous MF-AT was prepared using a Lipogems^®^ device (Lipogems International SpA, Milan, Italy). A Lipogems^®^ device is a device specifically used for liposuction and adipose tissue treatment. It can obtain pure autologous MF-AT through physical manipulation [13]. In this way, the product containing pericytes is retained within an intact stromal vascular niche and interacts with the recipient tissue after transplantation, thereby becoming activated as MSCs [14,15]. All patients were operated on in a randomized sequence.

After local anesthesia and sterilization were successfully induced, the swelling solution (composed of 500 mL normal saline, 50 mL lidocaine (2%) and 1 mL adrenaline (1:1000)) was injected into the extraction site. The gas in the device was emptied and the clamps at both ends of the device were closed. Then, liposuction was carried out at the premarked area with a vacuum syringe. The amount of adipose tissue mixture was generally 80–100 mL (Figure 1). The wound was covered with a dressing and the abdominal band was bandaged and fixed for 3 days. The next step is to manipulate the adipose tissue. Adipose tissue was injected into the device from the inlet, the outlet clamp was opened and 1/3 of the volume of the device was injected. The inlet clamp was opened, and flushing began. The clamps were closed at both ends, and vibration was performed for 30 s. The above operation was then repeated 4 times. During the washing process, lidocaine, blood, grease, etc., were removed and the autologous MF-AT was obtained. After washing, the clamp was closed at both ends, the device was turned upside down, a 10-mL syringe was connected to both ends of the device and the clamp was opened at the inlet. The syringe was pushed to obtain the autologous MF-AT. The autologous MF-AT was then injected into the knee joint. The patient should flex and extend the knee joint to ensure that the MF-AT are evenly distributed within the joint.

The autologous MF-AT were extracted intraoperatively and injected rapidly into the patient, so we did not quantitatively count the AD-MSCs injected into the patient but controlled for the volume of the fragment. After removing other impurities through the Lipogems^®^ device, the injection volume of all patients in this study was 6–8 mL per knee joint. The entire procedure lasted approximately 30 min. Because the damage caused by this surgery was very small and it was strictly performed aseptically, we did not use prophylactic antibiotics to prevent infection. One day after the operation, if there was no discomfort, the patient was discharged from the hospital.

### 2.4. Outcome Measures

Basic information such as patient age, sex, height, weight, body mass index (BMI), disease duration and preoperative comorbidities (e.g., diabetes, hypertension) were recorded. The Western Ontario and McMaster Universities (WOMAC) score, hospital for special surgery (HSS) score, knee society score (KSS) and visual analogue scale (VAS) pain score were recorded preoperatively 1 month, 3 months, 6 months, 9 months, 12 months and 18 months after the operation. All data were recorded by the same researcher who did not participate in the operation. See Figure 2 for details.

The degree of cartilage injury under magnetic resonance imaging evaluation (MRI) was described by Recht grading [16]. MRI was performed using a 3.0-T scanner with an 8-channel knee coil. The maximum gradient strength was 80 mT/m, while the maximum slew rate was 100 mT/m/ms. The thickness and spacing of the scanning layer were 3.0 mm and 0.6 mm, respectively. The images were digitally transmitted to the image archiving and communication system. Radiological measurements were made using the electronic calipers and goniometers provided with the software. The evaluations were performed by a senior radiologist who was blinded to the patient’s information.

Patients were considered to have postejection adverse events when they experienced hypersensitivity, fever, nausea and vomiting, cardiovascular events, severe pain, bleeding, marked swelling, surgical site infection and infection of the knee joint.

### 2.5. Statistical Analysis

The sample size was performed with PASS 2011 (NCSS, LLC, Kaysville, UT, USA) and calculated using WOMAC as the primary outcome. A difference of 10 points was assumed during the follow-up period (α = 0.05; β = 0.80). Considering an intra-group standard deviation of 15 points, 16 subjects were needed to obtain the required statistical power. Considering a 15% dropout rate, the minimum sample size was 19 cases. Statistical analysis was performed with SPSS version 26 (IBM, New York, NY, USA) and GraphPad Prism version 8 (GraphPad Software, San Diego, CA, USA). Continuous variables are subject to normal distribution based on the Shapiro–Wilk test. Continuous variables were reported as means and standard errors. Paired sample t-tests were used to test if there was a statistically significant difference between the two groups. Categorical variables were recorded as incidence and rate. All statistical tests were performed with bilateral tests, and *p* values less than or equal to 0.05 were considered statistically significant.

## 3. Results

### 3.1. Demographic Characteristics

A total of 20 patients with knee OA (40 knees) were included in this experiment. The 20 patients comprised 12 females and 8 males aged 54.63 ± 3.90 years (49–60 years), with a mean BMI of approximately 25.5 ± 2.86 kg/m^2^ (21.3–28.76 kg/m_2_). All of them had a history of knee OA for more than 2 years (6.9 ± 3.2 years) and received conservative treatment without surgery during this period, and the degree of lower limb coronal alignment was 12.58 ± 3.59° (5.5–18.0°). Seven patients had hypertension and four had type II diabetes mellitus. The remaining patients had no concomitant chronic diseases. See Table 1 for details.

### 3.2. Knee Function Outcomes

#### 3.2.1. WOMAC Score

At the last follow-up, the WOMAC scores of 20 patients after injection were improved (*p* < 0.05) from 24.87 ± 1.44 before injection to 17.13 ± 12.33 18 months after injection. The WOMAC score of 20 patients decreased continuously from before injection to 3 months, 6 months and 9 months after injection (*p* < 0.05). The scores at 9 months and 12 months after injection were compared (*p* > 0.05) (Table 2 (A and B)), and the results indicated that there was no significant difference between 9 months and 12 months after injection. From 12 months to 18 months after injection, the score began to increase (from 7.93 ± 6.44 to 17.13 ± 12.33, as shown in Figure 3 and Table 2 (A and B)), and there was a significant difference between the two scores (*p* < 0.05). These data indicate that the 20 patients had the best knee function at 9–12 months after the application of MF-AT. See Table 2 and Figure 3 for details.

#### 3.2.2. KSS and HSS Score

After 18 months of follow-up, the KSS of the patients increased from 77.73 ± 11.71 to 90.6 ± 6.49, and the HSS score increased from 79.73 ± 8.91 to 89.07 ± 6.98. There were significant differences between the scores before injection and 18 months after injection (*p* < 0.05, as shown in Table 2 (E–H) and Figure 3). The KSS of the 20 patients increased continuously from before injection to 3 months, 6 months and 9 months after injection (from 77.73 ± 11.71 to 95.93 ± 4.54), as did the HSS score (from 79.73 ± 8.91 to 95.93 ± 4.27). From 9 months to 12 and 18 months after injection, the scores of the two groups gradually decreased (KSS from 95.93 ± 4.54 to 90.6 ± 6.49, HSS score from 95.93 ± 4.27 to 89.07 ± 6.98, as shown in Table 2 (E and G) and Figure 3), and the scores at each follow-up were worse than those at the last follow-up (*p* < 0.05). See Table 2 and Figure 3 for details.

#### 3.2.3. VAS Pain Score

From before injection to 3 months, 6 months and 9 months after injection, the VAS pain score of the patients decreased continuously (from 4.2 ± 1.42 to 0.4 ± 0.63), and there was a significant difference from the score before injection (*p* < 0.05, as shown in Table 2 (C and D) and Figure 3). The VAS pain score gradually increased from 9 months to 12 and 18 months after injection (from 0.4 ± 0.63 to 1.73 ± 0. 88, *p* < 0.05). However, 18 months after injection compared with before injection, the score was improved, and there was a significant difference (*p* < 0.05), indicating that 18 months after injection, the patient’s pain symptoms were relieved and that the pain symptoms were the mildest and the effect was the best in the ninth month after the application of MF-AT. See Table 2 and Figure 3 for details.

### 3.3. Radiological Outcomes 

Twenty-eight of forty knees were Recht grade II, and twelve were Recht grade III before injection. Eighteen months after injection, eight were Recht grade I, twenty-two were Recht grade II, and ten were Recht grade III. The Recht score improvement rate was 25%. In Appendix A, the MRI findings of two patients before and after receiving an intra-articular injection of MF-AT are enumerated. In the coronal and sagittal planes, we could see that the knee oedema was slightly reduced compared with that before the application of MF-AT, but cartilage regeneration was not significant.

### 3.4. Safety

There were no hypersensitivity, fever, nausea and vomiting, cardiovascular events, severe pain, bleeding, marked swelling, surgical site infection and infection of the knee joint during follow-up. All patients were discharged on the first day after injection.

After the injection of autologous micro-fragmented adipose tissue, knee function gradually improved with the platform period of 9–12 months, and the effect began to decline after 12 months. Score. HSS: hospital for special surgery score; KSS: knee society score; WOMAC: the Western Ontario and McMaster Universities; VAS: visual analogue scale.

## 4. Discussion

The most important finding of this study is that a single intra-articular injection of MF-AT in patients with knee OA resulted in satisfactory clinical results and functional improvement with no adverse events at the 18-month follow-up, with patients showing the best improvement in knee function by 9 months post-injection.

Knee OA is a chronic progressive degenerative disease that can lead to knee pain and dysfunction and seriously affect daily life, especially in elderly individuals. Pain and loss of function are the main clinical features leading to treatment, including nonpharmacological, pharmacological, and surgical methods [17]. The development and progression of OA is a multifactorial disease, and patients vary greatly in age, BMI and daily activities [18,19,20]. To date, no treatment can reverse knee degeneration or promote articular cartilage regeneration. However, in the body, MSCs are found mainly in tissues such as the periosteum, synovium, adipose tissue and bone marrow and are suitable cells for repairing damaged tissues; under certain conditions, they can efficiently differentiate into muscle, fat and cartilage and other tissues [11]. Some studies have shown that after intra-articular injection, MSCs attach to cartilage defects, increase in number and participate in the regeneration of articular cartilage [8,21]. Adipose tissue is an easily accessible source of MSCs, and its micro fractionation state MF-AT allows for rapid harvesting of relevant volumes of minimally manipulated tissue consisting of clusters containing MSCs [22]. Furthermore, compared to raw adipose tissue, MF-AT contains fewer leukocytes and supra-adventitial-adipose stromal cells, as well as abundant endothelial progenitor cells, which have been described to maintain proliferation and differentiation in interaction with tissue-resident cells [23].

MSCs are depleted, and their ability to proliferate and differentiate is reduced in the microenvironment of osteoarthritis [24]. Therefore, providing large quantities of healthy and functional MSCs helps promote repair or inhibits the progression of cartilage loss [25]. Adipose tissue is much less expensive and invasive to obtain than bone marrow and is more abundant. In addition, the proliferation rate of MSCs from adipose tissue is higher than that derived from of bone marrow [26]. Lipogems^®^ is a simple system designed to collect, process and transfer refined adipose tissue, with great regenerative potential and optimal processing capacity [13]. The adipose tissue is mechanically shredded and washed until there is no free proinflammatory oil or blood residue, and the resulting product is composed of small intact adipose tissue clusters (250–650 microns) containing pericytes retained in intact interstitial vascular niches [13,27]. The end product MF-AT is injected into the joint cavity and activated as MSCs and begins its regeneration process. Therefore, this product is more widely used clinically. However, clinicians and patients remain concerned about the safety of the product. Lipogems^®^ technology obtains autologous MF-AT after special treatment with autologous fat, which can restore the microenvironment of cell growth to the maximum extent. In theory, these microparticles have been considered safe since they do not lead to the formation of antibodies [28]. There may be a decreased risk of tumorigenesis, disease transmission and host immune rejection compared with the use of allogeneic adipose-derived MSCs [29]. However, in previous studies using allogeneic MSCs for intra-articular injection, few adverse events were reported, and the improvement of clinical symptoms was not greatly different from studies using autologous MSCs [30,31]. Intra-articular injectable allogeneic MSCs still need to undergo further clinical study, including for clinical efficacy and safety. Davatchi et al. conducted a study in 2016 on three patients who were injected with autogenous bone marrow MSCs and followed up for 5 years [32]. The indicators of these three patients significantly improved and then gradually deteriorated six months after injection. However, the knee function was still better than that before injection, and no adverse events occurred in the fifth year. Our results show similar functional changes and we conclude that autologous MF-AT is safe for short-term injection. The follow-up period of this study was 18 months, and no serious adverse events occurred during this time.

After the injection of autologous MF-AT, the expected effect appeared at 3–6 months, the platform period was 9–12 months, and the effect began to decline after 12 months. The patients showed the best improvement in knee function by 9 months post-injection in this study. These data all show that after a single injection of MF-AT, the effect can be maintained for approximately 9 months, with some improvement in patients’ pain symptoms and knee function. Therefore, intra-articular injection of autologous MF-AT may be a viable option for the treatment of degenerative osteoarthritis. In addition, there are case reports showing that patients with OA and unresponsive pain in the knee joint associated with meniscal injury treated with intra-articular injections of AD-MSCs have improved joint function [33]. Recently, intra-articular injection of AD-MSCs in one patient with posttraumatic cartilage injury improved knee function [34]. Twelve weeks after injection of AD-MSCs, the Oxford knee score improved from 36 (baseline) to 46, and MRI studies 12 months after injection demonstrated an improved cartilage damage signal. Thirty months after receiving an injection, the patient was able to ski naturally. In the present study, we injected one autologous MF-AT, a blend, within the articular cavity, which was comparable to other studies that independently used MSCs [9,30,35,36] and added a rich cell population, intact three-dimensional bioscaffolds and multiple cell growth factors [8,29,34]. These adjuvants may better sustain the repair of MSCs, relieve pain and improve joint function in patients.

By following up with patients and performing MRI examinations, we were able to intuitively observe the effects of autologous MF-AT on knee cartilage before and after injection at different stages. In several previous studies [10,30,37], similar cartilage changes after intra-articular injection of MSCs in patients with knee OA were reported. These studies measured cartilage by T2 relaxation time and cartilage index. Lee et al. conducted a prospective, double-blind, randomized, controlled phase IIb clinical trial in 2019 in which 12 patients (experimental group) were given AD-MSCs and compared with 12 patients (control group) who received normal saline [11]. The observation time was 6 months, and no obvious changes in cartilage were found in the experimental group at the 6-month follow-up after injection; however, the cartilage defects were significantly increased in the control group. Since the results for the measurement of cartilage volume on MRI are inconclusive internationally, in this study, we judged cartilage changes only by observing the MRI images of the patients, and we did not quantify the cartilage changes. Currently, the T2 mapping technique is considered to have great potential for use in the clinic [38]. The T2 mapping technique can assess cartilage structural integrity, tissue structure and water content by measuring T2 values, which is beneficial for the sensitive measurement of histological changes in cartilage. However, this technique has high sensitivity and low specificity, and the measurement of T2 values is performed manually by clinicians, making it susceptible to subjective effects and unfavorable for early diagnosis. Therefore, T2 mapping sequences have not been widely used in the clinical diagnosis of articular cartilage imaging [39]. In the MRI images before and after the injection, we could observe a partial improvement in the Recht grading of the knee cartilage, but there was no significant evidence of cartilage regeneration. Over time, the effect of autologous MF-AT on cartilage still needs further study.

Our study still had some limitations. First, the sample size of the experiment was small, with only 20 patients (40 knees) participating in the present experiment, and specific individuals had a large effect on the experiment. Second, conclusions regarding efficacy are weak due to the lack of a control group and an inability to compare our patients with patients who did not undergo MF-AT. Third, since MF-AT is a mixed compound, with so many active components, would the activity and effect vary among different people, especially given the differences in age, BMI and daily activity, among other variables. Moreover, autologous MF-AT was extracted intraoperatively and rapidly injected into the patients, so we did not quantify the AD-MSCs injected into the patients. In addition, observation of cartilage regeneration by conventional MRI alone may be inadequate, and methods are needed that can quantitatively assess the volume of cartilage regeneration. Despite these shortcomings, given the limited data on this particular treatment method, we believe this study is a valuable addition to the literature.

## 5. Conclusions

It has been found that intra-articular injection of autologous MF-AT provided satisfactory functional improvement and pain relief in patients with knee OA and caused no adverse events at the 18-month follow-up. However, the improvement in knee function did not persist, with optimal results occurring 9–12 months after injection, and cartilage regeneration remaining to be discussed. Future prospective multicenter studies with large samples are needed to evaluate the long-term effects of autologous MF-AT in patients with knee OA.

## Figures and Tables

**Figure 1 jpm-13-00504-f001:**
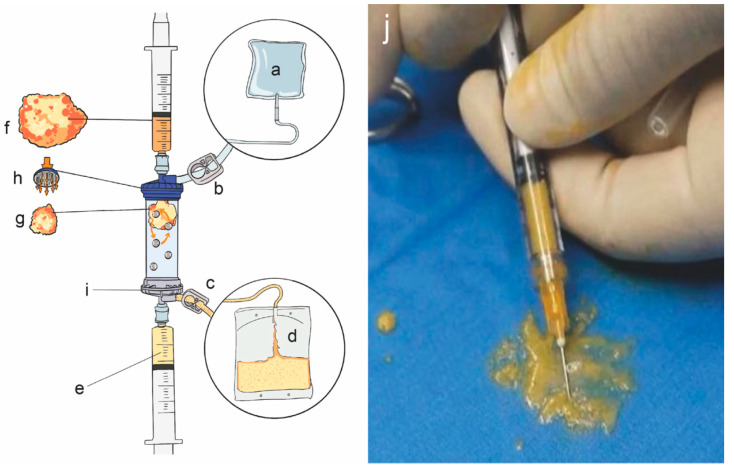
Lipogems^®^ device and autologous micro-fragmented adipose tissue. (**a**): Physiological saline bag (3–5 L); (**b**): inlet clamp; (**c**): outlet clamp; (**d**): waste bag: collection of oils and blood residues; (**e**): final products; (**f**): pristine adipose tissue; (**g**): adipose tissue after shaking and emulsification; (**h**): primitive adipose tissue entry; (**i**): final product collection port; (**j**): autologous micro-fragmented adipose tissue.

**Figure 2 jpm-13-00504-f002:**
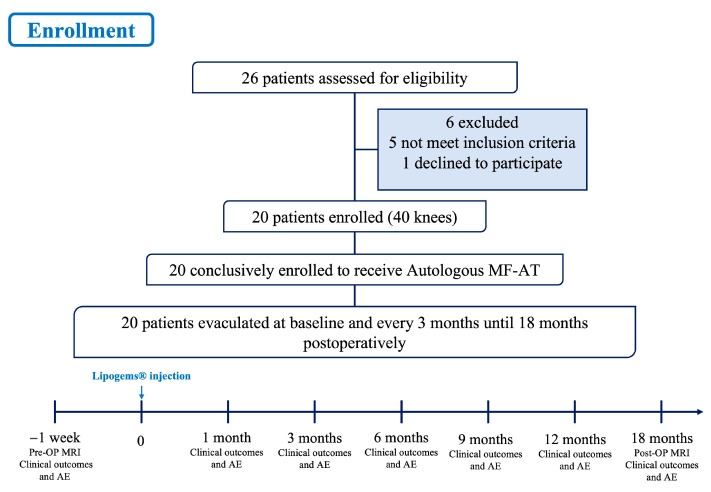
Flow chart of the clinical trial. MF-AT: micro-fragmented adipose tissue; AE: adverse events.

**Figure 3 jpm-13-00504-f003:**
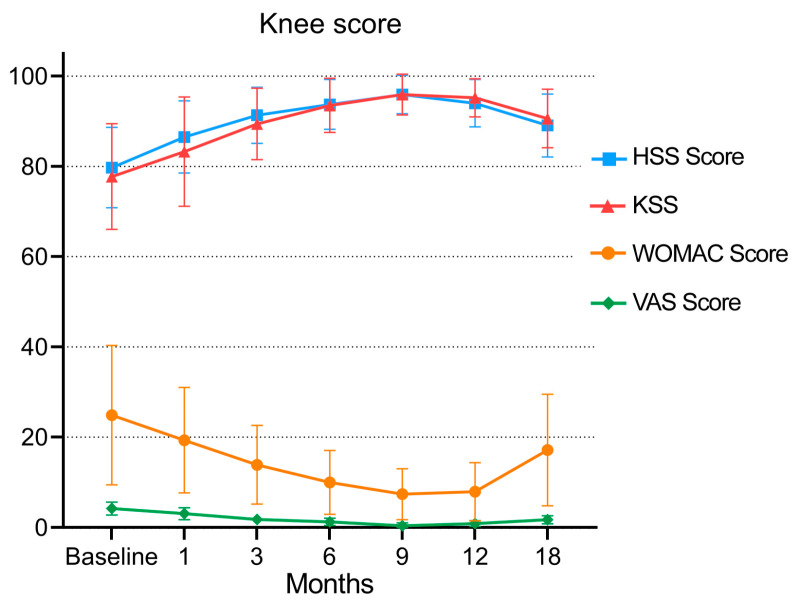
Changes in the WOMAC score, VAS, HSS score and KSS during follow-up.

**Table 1 jpm-13-00504-t001:** Demographic characteristics of the patients.

	Values
Patients (n)	20
Knees (n)	40
Age (years)	55 ± 3.9
Sex (n, %)	
Male	8 (40%)
Female	12 (60%)
Height (cm)	165 ± 0.05
Weight (kg)	69.3 ± 8.48
Body mass index (kg/m^2^)	25.5 ± 2.86
Symptom duration (years)	6.9 ± 3.2
Lower limb alignment (varus angle°)	12.58 ± 3.59
Baseline knee function	
WOMAC score	24.87 ± 15.44
VAS	4.20 ± 1.42
KSS	77.73 ± 11.71
HSS score	79.73 ± 8.91
Comorbidities	AHT in 7 patients (35%) DM in 4 patients (20%)

Abbreviations: WOMAC: the Western Ontario and McMaster Universities Osteoarthritis Index; VAS: visual analogue score; KSS: knee society score; HSS: hospital for special surgery; AHT: arterial hypertension; DM: diabetes mellitus.

**Table 2 jpm-13-00504-t002:** Changes in knee joint function outcomes during follow-up.

**A. WOMAC Score Comparison between Adjacent Follow-Up Periods**							
**WOMAC score**	**Baseline**	**1 month**	**3 months**	**6 months**	**9 months**	**12 months**	**18 months**
Mean	24.87	19.33	13.87	10.00	7.40	7.93	17.13
SD	15.44	11.68	8.70	7.04	5.63	6.44	12.33
T		3.517	4.323	6.526	4.516	−0.725	−4.545
*p* value		0.003	0.001	0.001	0.001	0.481	0.001
**B. WOMAC Score Comparison between Specific Time Periods**							
**WOMAC score**		**Mean**		**SD**		**T**	***p* value**
Baseline		24.87		15.44		6.380	0.001
9 months		7.40		5.63			
Baseline		24.87		15.44		5.528	0.001
18 months		17.13		12.33			
9 months		7.40		5.63		−4.935	0.001
18 months		17.13		12.33			
**C. VAS Comparison between Adjacent Follow-Up Periods**							
**VAS**	**Baseline**	**1 month**	**3 months**	**6 months**	**9 months**	**12 months**	**18 months**
Mean	4.2	3.07	1.8	1.27	0.4	0.87	1.73
SD	1.42	1.33	1.08	0.8	0.63	0.64	0.88
T		5.906	6.141	2.779	5.245	−2.432	−4.516
*p* value		0.001	0.001	0.015	0.001	0.029	0.001
**D. VAS Comparison between Specific Time Periods**							
**VAS**		**Mean**		**SD**		**T**	***p* value**
Baseline		4.2		1.42		12.192	0.001
9 months		0.4		0.63			
Baseline		4.2		1.42		9.012	0.001
18 months		1.73		0.88			
9 months		0.4		0.63		−8.367	0.001
18 months		1.73		0.88			
**E. KSS Comparison between Adjacent Follow-Up Periods**							
**KSS**	**Baseline**	**1 month**	**3 months**	**6 months**	**9 months**	**12 months**	**18 months**
Mean	77.73	83.27	89.4	93.53	95.93	95.2	90. 6
SD	11.71	12.12	7.9	6.03	4.54	4.23	6.49
T		−8.98	−3.992	−5.141	−3.032	1.661	6.2
*p* value		0.001	0.001	0.001	0.009	0.119	0.001
**F. KSS Comparison between Specific Time Periods**							
**KSS**		**Mean**		**SD**		**T**	***p* value**
Baseline		77.73		11.71		−8.684	0.001
9 months		95.93		4.54			
Baseline		77.73		11.71		−6.940	0.001
18 months		90.6		6.49			
9 months		95.93		4.54		6.904	0.001
18 months		90.6		6.49			
**G. HSS Score Comparison between Adjacent Follow-Up Periods**							
**HSS score**	**Baseline**	**1 month**	**3 months**	**6 months**	**9 months**	**12 months**	**18 months**
Mean	79.73	86.53	91.33	93.73	95.93	94	89.07
SD	8.91	8.02	6.22	5.52	4.27	5.26	6.98
T		−6.961	−6.339	−4.505	−4.785	2.377	6.396
*p* value		0.001	0.001	0.001	0.001	0.032	0.001
**H. HSS Score Comparison between Specific Time Periods**							
**HSS score**		**Mean**		**SD**		**T**	***p* value**
Baseline		79.73		8.91		−9.350	0.001
9 months		95.93		4.27			
Baseline		79.73		8.91		−5.709	0.001
18 months		89.07		6.98			
9 months		95.93		4.27		6.198	0.001
18 months		89.07		6.98			

Abbreviations: WOMAC: the Western Ontario and McMaster Universities; VAS: visual analogue scale; KSS: knee society score; HSS: hospital for special surgery score; SD: standard errors.

## Data Availability

The data associated with the paper are not publicly available but are available from the corresponding author upon reasonable request.

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
