# Peer review of "Intra-Articular Injection of Autologous Micro-Fragmented Adipose Tissue for the Treatment of Knee Osteoarthritis: A Prospective Interventional Study"

_jpm, 2023, doi:10.3390/jpm13030504_

Round 1
Reviewer 1 Report
1. 2.1. Study design: “The study is a single-center, interventional, prospective, observational study”. The study is interventional, not observational. Of course, one observes the effects of an intervention, but study design classification is different.
2. 2.1. Study design: “A total of 20 patients with knee OA participated in the experiment.” This is a result of the study, and not a design characteristic, since no one really fixes a priori the number of participants in a study. Therefore, I would recommend moving this information to the results section.
3. 2.2 Patient selection. Authors have stated that the study included patients “in good physical condition”. One understands this phrase in a non-academical scenario, but in a scientific research article one must be more precise. This is especially important since one learns form the results section that some patients had hypertension and others had diabetes.
4. 2.2 Patient selection. Authors have stated that the study included patients “able to tolerate surgery”. This sounds like what two anesthesiologists would say on a lunch break. What does it mean specifically? How did the authors determine that those patients were “able to tolerate surgery” before surgery?
5. 2.2 Patient selection. The authors state that they exclude patients with “rheumatoid diseases”. There is no such thing as “rheumatoid disease”, which would literally mean diseases which look like rheumatoid arthritis (RA). There is “RA” and there are “RHEUMATIC diseases” – which one did the authors use for exclusion, patients with RA and/or patients with other rheumatic diseases such as psoriatic arthritis, spondyloarthritis, gout etc.?
6. 2.2 Patient selection. There is no information on actual patient selection process. For example, where did the 26 screened patients come from (Where they already admitted to the hospital? Were they outpatients? Why did they come to the hospital?)? When were the patients included (a timeframe should be given)? Were the patients included randomly and, if yes, how did the authors achieve randomness?
7. 2.2 Patient selection. Were the patients on any drugs for KOA (NSAIDs, opioids, paracetamol, SYSADOA)?
8. Statistical analysis. The authors state that “Continuous variables were recorded as means and standard errors.” Firstly, these variables were not recorded, but reported: “continuous variables were REPORTED as means and standard errors”. Secondly, by reporting them as such and by analyzing them with t tests, one might understand that these variables were normally distributed? Where they normally distributed? How did the authors assess normality of distribution?
9. Table 1 reports KL grades. Were knee radiographs taken? The methods section does not mention knee radiographs. Also, KL should be explicit, at least in the footnote of the table (KL – Kellgren Lawrence), as well as the other abbreviations, since tables should be able to stand alone from the text. More so, KL classification should have a reference/citation.
10. Table 1 reports that “Seven patients had hypertension and four had type II diabetes mellitus”. Please use numbers. Please add 2 more lines to the table, one named arterial hypertension (AHT) with the value 7 (35%) and another one named diabetes mellitus (DM) with the value 4 (20%).
11. Conclusions. Please move this phrase “Despite these shortcomings, given the limited data on this particular treatment 355 method, we believe this study is a valuable addition to the literature.” from the beginning of the conclusions to the end of the discussion section.
12. Reference 2, 33, 34 are incomplete.
Author Response
Please see the file we uploaded.

Reviewer 2 Report
The authors have conducted a prospective study on the application of cellular lipid fragments for the treatment of patients with osteoarthritis. Although the approach is not novel, since the instrumentation is previously patented and used by other authors in the literature, the results provided may corroborate those previously obtained by other authors and expand the data and knowledge of this type of treatment.
Some corrections and suggestions on the content of the text are as follows:
- Line 56-57: The authors claim a lack of studies addressing this methodology. There are numerous references in the literature to the application of adipose fragments for the treatment of knee OA (PMID: 31861180; PMID: 31328447; PMID: 31571571; etc.). In fact, there are numerous clinical trials applying this method (https://clinicaltrials.gov/ct2/results?cond=Osteoarthritis%2C+Knee&term=adipose%2C+fragmented&cntry=&state=&city=&dist=). The authors should mention the non-novelty of their hypothesis. It is also strange when, in the discussion section, the authors compare their results with those of other researchers performing similar methodologies.
-Line 106: The authors allege the impossibility of counting the cells before implantation in the patient. Are there no bibliographic references that refer to a possible average cell quantity per extraction? Since this is a minimally invasive process, it would be very useful if the authors could perform extractions of adipose fractions to standardize the average cellular quantity per process.
-Line 130: improve the wording of the sentence.
-Line 209: put quantity in numbers as previously in the text.
-Table 2. The figure caption should be more explanatory. The abbreviations SD, T and P are missing. If there are comparisons between groups, this should also be explained in the figure caption.
-Line 267. What do you mean with "epithelial layer adipose stromal cells"? The reference supplied does not appear that term. In fact, epithelial tissues are composed by epithelial cells, being adipocytes mainly located immersed in connective tissues.
-Line 267: improved wording of the sentence. sults in terms of cartilage regeneration? Justify the statement. Cartilage tissue is an avascular tissue and the formation of vessels in it produces pathologic conditions.
-Line 273-275. improve sentence wording.
-Line 286-288: The authors make reference to the fact that the use of autologous MF-ATs may increase the risk of disease transmission and immune response compared to allogeneic MSCs. It is strange that the patient's own cells are more likely to generate these harms than donor cells. Please check the meaning of this sentence.
Finally, due to tissue affected during OA, the possible analysis of the images obtained by MNR or ultrasound would have yielded interesting quantitative and qualitative data on the possible effect and reaction of the MF-ATs in the lesion area, beyond other values, of a more qualitative nature, analyzed by the authors.
